# Resisting Over-Smoothing in Graph Neural Networks via Dual-Dimensional Decoupling

Wei Shen
National Engineering Research
Center for Multimedia Software,
School of Computer Science, Wuhan
University
Wuhan, China
weishen@whu.edu.cn

Mang Ye*
National Engineering Research
Center for Multimedia Software,
School of Computer Science, Wuhan
University
Wuhan, China
yemang@whu.edu.cn

Wenke Huang
National Engineering Research
Center for Multimedia Software,
School of Computer Science, Wuhan
University
Wuhan, China
wenkehuang@whu.edu.cn

## Abstract

Graph Neural Networks (GNNs) are widely employed to derive meaningful node representations from graphs. Despite their success, deep GNNs frequently grapple with the oversmoothing issue, where node representations become highly indistinguishable due to repeated aggregations. In this work, we consider the oversmoothing issue from two aspects of the node embedding space: dimension and instance. Specifically, while existing methods primarily concentrate on instance-level node relations to mitigate oversmoothing, we propose to mitigate oversmoothing at dimension level. We reveal the heightened information redundancy between dimensions which diminishes information diversity and impairs node differentiation in GNNs. Motivated by this insight, we propose Dimension-Level Decoupling (DLD) to reduce dimensional information redundancy, enhancing dimensional-level node differentiation. Besides, at the instance level, the neglect of class differences leads to vague classification boundaries. Hence, we introduce Instance-Level Class-Difference Decoupling (ICDD) that repels inter-class nodes and attracts intra-class nodes, improving the instance-level node discrimination with clear classification boundaries. Additionally, we introduce a novel evaluation metric that considers the impact of class differences on node distances, facilitating precise oversmoothing measurement. Extensive experiments demonstrate the effectiveness of our method Dual-Dimensional Class-Difference Decoupling (DDCD) across diverse scenarios. Codes are available at https://github.com/shentt67/DDCD.

## CCS Concepts

• **Mathematics of computing** → **Graph algorithms**; • **Computing methodologies** → **Regularization**.

## Keywords

Graph Neural Networks, Oversmoothing, Regularization

*Corresponding author

**ACM Reference Format:**
Wei Shen, Mang Ye, and Wenke Huang. 2024. Resisting Over-Smoothing in Graph Neural Networks via Dual-Dimensional Decoupling. In *Proceedings of the 32nd ACM International Conference on Multimedia (MM '24), October 28-November 1, 2024, Melbourne, VIC, Australia.* ACM, New York, NY, USA, 10 pages. https://doi.org/10.1145/3664647.3681204

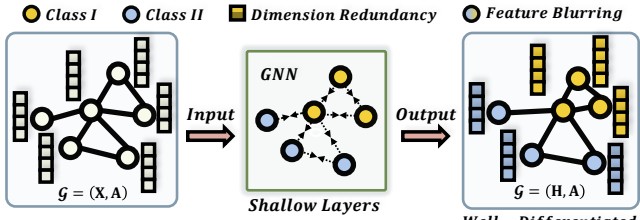

(a) Well-differentiated nodes with shallow GNN layers.

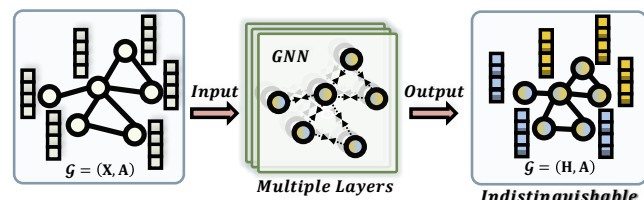

(b) Indistinguishable nodes with multiple aggregations.

**Figure 1: Illustration of *Oversmoothing*. (a) With shallow GNN layers, the node representations are well-differentiated. (b) The repeated information aggregations in deep GNNs lead to highly indistinguishable node representations. This phenomenon is widely recognized as *oversmoothing* issue.**

## 1 Introduction

Graph data encodes information on both individual entities and the relationships between those entities, finding wide applications in practical scenarios such as social recommendation [14, 49, 81], node classification [26, 29, 32, 59], biology design [45, 51], and so on [43, 48, 58, 60, 68, 71]. To extract meaningful information from graph data, Graph Neural Networks (GNNs) [12, 32, 57, 61, 65, 82] are widely adopted and have shown promising results in generating effective node representations. Inspired by the success of deepening Convolutional Neural Network (CNN) layers [21, 33, 54, 55], recent works have attempted to improve GNN performance by increasing network depth to mine the multi-hop relation of graph

data. However, deeper GNN layers often suffer from performance degradation, which has been the subject of extensive discussions in recent years. Li *et al.* [38] first revealed that repeated information aggregations in deep GNNs lead to highly indistinguishable node representations and hampers performance. This phenomenon is widely recognized as *oversmoothing* issue [2, 30, 38, 46, 66].

Numerous efforts have been made to alleviate oversmoothing [27, 35, 50, 78, 79]. Several works tackle oversmoothing by modifying the model architecture. Li *et al.* [35, 36] proposed to apply residual/dense connections and achieve better performance with deeper GNNs. Chen *et al.* [44] proposed to utilize identity mapping to improve GNN capabilities. Zhou *et al.* [80] investigated oversmoothing by the Dirichlet energy and developed a generalized principle to train deep GNNs. However, the designs on model architecture are inflexible and introduce additional network parameters, limiting their applications. Besides, several works attempt to address oversmoothing from the graph structure and node embeddings, which can be appropriately applied to existing deep GNNs: Dropout-based methods [15, 42, 50] randomly drop node features or graph edges to avoid redundant aggregation but can inadvertently disrupt the node and structural information of the graph, losing original graph information; Normalization-based methods [27, 78, 79] address oversmoothing by normalizing node representations after each layer to preserve node distinctiveness, are versatile and can be readily applied to various GNN models. However, they mainly focus on instance-level cues and the dimension-level relations for enhancing node differentiation remain unexplored.

Beyond the aforementioned limitations, we consider the oversmoothing issue from two perspectives of the node embedding space: dimension and instance. In self-supervised learning, dimensional relations have been demonstrated to improve the representation capacity of the embedding space [1, 37, 39, 75]. Likewise, we assume that dimensional relations can also enhance the node representation in GNNs. We empirically uncovered that the pronounced information redundancy between dimensions hinders the capability of embeddings and impacts the node differentiation ability in GNNs (detailed in Section 3.3). Drawing inspiration from these insights, we propose to enhance node differentiation at the dimension level by reducing the information redundancy between dimensions of node embeddings in deep GNNs. To achieve this, we introduce the **Dimension-Level Decoupling (DLD)** that minimizes the covariance between dimensions in each GNN layer. It reduces the redundant information within node embedding dimensions, which effectively promotes diversity dimension information and enhances the differentiation ability of node embeddings.

Besides, existing instance-level methods normalize node representations without taking class differences into account, resulting in vague classification boundaries and performance degradation. For instance, Zhou *et al.* [79] introduced Differentiable Group Normalization (DGN) that assigns group probabilities and normalizes node representations based on group relations. However, nodes from different classes can be grouped together, leading to fuzzy class boundaries. To utilize class differences for explicit classification boundaries, we introduce the **Instance-Level Class-Difference Decoupling (ICDD)** to alleviate the oversmoothing issue, which can be applied to various GNN architectures. The basic idea is to repel the inter-class nodes and attract intra-class nodes after each

GNN layer, dynamically encouraging inter-class nodes to separate while maintaining similarity among intra-class node features. By decoupling node representations based on class differences, this strategy substantially enhances instance-level differentiation with explicit classification boundaries.

Additionally, existing metrics [6, 23] assess oversmoothing based on the distances between node pairs. However, the pair-wise distances are associated with class information: nodes in different classes should exhibit distinct pair-wise distances while having closer pair-wise distances in the same class. Without considering class differences, the pair-wise distances are affected by class information and fail to accurately assess oversmoothing. In this case, we introduce a new metric: **R**elative **I**nter-**C**lass **D**istance **(RICD)**, which calculates the similarity difference between inter-class nodes and intra-class nodes. It focuses on the relative difference between inter-class nodes and intra-class nodes, accounting for the influence of class information on the accurate assessment of oversmoothing. In summary, our primary contributions can be outlined as follows:

- We propose a dual-dimensional decoupling method for alleviating oversmoothing, which can be readily applied to various deep GNNs. Firstly, at the dimension level, we propose to reduce dimension redundancy by minimizing dimension covariances, diversifying the dimension information and ultimately promoting well-differentiated nodes for mitigating oversmoothing.

- We propose incorporating class differences at the instance level to alleviate oversmoothing with explicit class boundaries. It dynamically encourages inter-class nodes to separate while maintaining similarity among intra-class node representations, significantly enhancing instance-level differentiation.

- We propose a novel metric to measure oversmoothing. It quantifies the similarity difference between inter-class nodes and intra-class nodes, considering the impact of class differences and accurately measuring the extent of oversmoothing.

- We empirically demonstrate our proposed method **DDCD (D**ual-**D**imensional **C**lass-**D**ifference **D**ecoupling), can effectively mitigate oversmoothing by combining dimension and instance level cues, achieving state-of-the-art compared to existing methods.

## 2 Related Work

### 2.1 Graph Neural Network

Graph Neural Networks (GNNs) [13, 17, 52, 76] are proposed to generate meaningful node representations by propagation and transformation over multiple network layers. They can be broadly categorized into spectral-based [3, 12, 22, 32] and spatial-based [16, 19, 57] models. Spectral methods perform convolution operations in the graph domain using spectral filters [3] and their simplified variants. In spatial GNNs, convolution operations are carried out by propagating and aggregating local information along the edges of a graph. Different aggregation functions are employed in spatial GNNs to learn node representations, including mean/max pooling [19], self-attention [57], and summation [64].

### 2.2 Oversmoothing

Oversmoothing [2, 30, 38, 46, 66] is first defined by Li *et al.* [38], where the node representations become highly indistinguishable

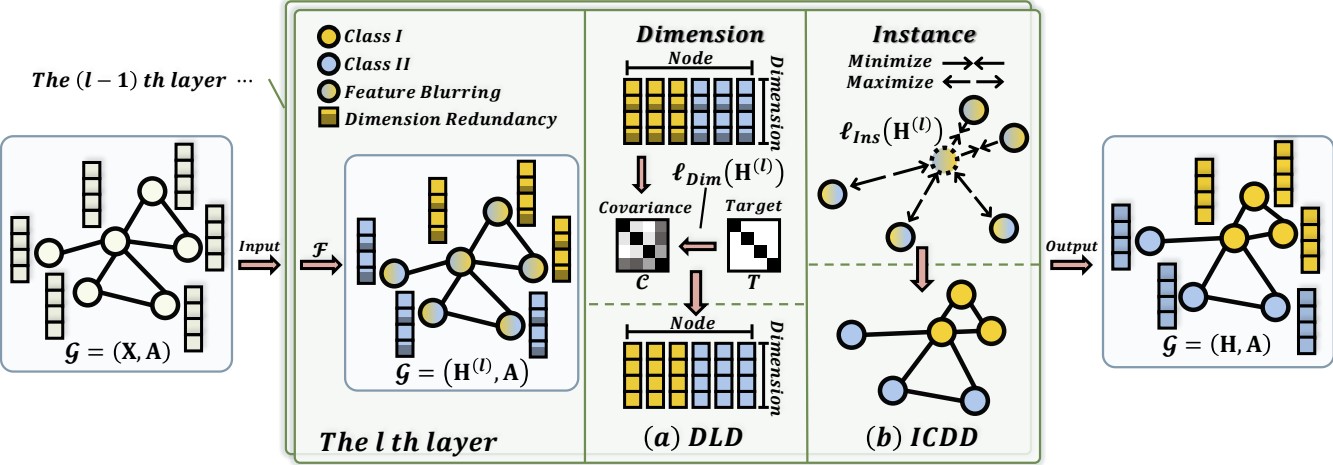

**Figure 2: Framework of Dual-Dimensional Class-Difference Decoupling (DDCD). To enhance the dimension-level discriminability of nodes, (a): Dimension-Level Decoupling (DLD in Section 3.3) urges diverse dimensions by minimizing the information redundancy between dimensions, bringing well-differentiated representations; To alleviate oversmoothing with clear class boundaries, (b): Instance-Level Class-Difference Decoupling (ICDD in Section 3.4) utilizes class differences by repelling nodes from different classes and attracting nodes from the same classes, which promotes distinguishable node representation in instance-level. As such, the output graph $\mathcal{G} = (H, A)$ has distinguishable node representations H. Best viewed in color.**

in deep GNNs. Li *et al.* [38] attributed this issue to the multiple Laplacian smoothing steps and suggested that oversmoothing was a common phenomenon in GNNs as the Laplace operator is a ubiquitous component, hindering the superposition of GNN layers. Recent research [2, 30, 62, 67] has explored oversmoothing in-depth, emphasizing its significance in developing deep GNNs to enhance node representations [7, 11, 77, 80]. Several works alleviate oversmoothing from the perspective of model architecture designs [18, 34, 35, 44, 80]. However, the designs on model architecture are inflexible and introduce additional network parameters, hindering their applications. Some approaches involve dropping nodes or edges to avoid redundant aggregations [15, 42, 50], but inadvertently disrupting the node and structural information of the graph, leading to the loss of original graph information and performance degradation. Other methods address oversmoothing through layer-wise normalization [27, 78, 79], but the ignorance of class differences results in vague class boundaries and ultimately hinders performance. In this work, we present a flexible regularization method addressing oversmoothing with class differences.

## 2.3 Dimensional Information Redundancy

Dimensional Information Redundancy [41, 74] indicates a high degree of redundant information encoded across these dimensions. It restricts the diversity of dimension information, thereby impeding the representation capability of feature embeddings. This phenomenon is prevalent and has been extensively investigated in self-supervised learning [1, 25, 28, 37, 39, 75]. In this work, we first propose to alleviate oversmoothing at the dimension level. We reveal the heightened dimension information redundancy in GNNs, which inspires us to enhance dimension diversity for node differentiation by reducing the redundancy between dimensions.

## 2.4 Contrastive Learning

Contrastive Learning is widely applied [5, 70, 72], which maximizes the agreement between positive samples while minimizing the agreement between negative samples. It utilizes augmentations to construct contrastive samples and is often applied in self-supervised learning for label-scarce scenes [4, 9, 20, 63, 73] or for pre-training networks [8, 10, 31, 47, 56, 70] to boost downstream tasks. In this work, we utilize class differences to construct contrastive samples, suppressing oversmoothing with clear class boundaries.

## 3 Methodology

### 3.1 Prelimilaries

We adopted the semi-supervised node classification (SSNC) setting proposed by Kipf and Welling [32] for our experiments. Specifically, in an undirected graph $\mathcal{G} = (\mathcal{V}, \mathcal{E}, X)$, each node $v_i$ in $\mathcal{V}$ is associated with a feature vector $x_i \in \mathbb{R}^d$, where $X = [x_1, ..., x_n]$ represents the initial feature matrix. $\mathcal{E} \subseteq \mathcal{V} \times \mathcal{V}$ denotes the edge set between nodes. The subset $\mathcal{V}_s \subseteq \mathcal{V}$ of nodes are labeled with $y_i \in \{0, ..., c-1\}$ for $v_i \in \mathcal{V}_s$, where $c$ is the number of classes. The adjacency matrix $A \in \{0, 1\}^{N \times N}$ encodes the relations between nodes, where $A_{ij} = 1$ indicates edge connections between node $v_i$ and $v_j$, otherwise $A_{ij} = 0$.

The task of SSNC is to predict the label of $x_i \notin \mathcal{V}_s$ using the labeled nodes and graph structure. In GNNs, multiple network layers are utilized for propagation and transformation, with a linear layer for predicting the labels of nodes. Denoting the transformation and propagation as $\mathcal{F}, \mathcal{P}$, the forward process of multi-layer GNNs can be formulated as:

$$H^{(l)} = \mathcal{F}(\mathcal{P}(H^{(l-1)}, A)). \tag{1}$$

The number of GNN layers is $n$, and $H^{(l)}$ represents the output of $l$-th GNN layer where $l \in \{1, ...n\}$.

## 3.2 Relative Inter-Class Distance

When deepening GNNs by repeatedly aggregating neighbor node representations, the node representations tend to become highly indistinguishable, i.e., the oversmoothing issue. A robust evaluation mechanism is critical for investigating this issue. Existing works [6, 23] measure the degree of oversmoothing based on the distance between node pairs. However, the pair-wise distances are closely associated with graph information (e.g., the class labels). For the node classification task, nodes in different classes should exhibit distinct pair-wise distances, while nodes in the same class should have closer pair-wise distances. Without the consideration of class information, the pair-wise distances are affected by class labels and fail to accurately assess oversmoothing. In this case, we propose a new metric: Relative Inter-Class Distance, which considers the impacts of class differences to accurately measure oversmoothing.

**Relative Inter-Class Distance (RICD).** Denote the class set in the graph is $\mathcal{M}$. For class $a \in \mathcal{M}$, the node set is denoted by $\mathcal{N}_a$, and the node feature vector in $\mathcal{N}_a$ is denoted as $n_a$. Denote $|\cdot|$ is the set cardinality. The cosine similarity of node feature vectors $n_a$ and $n_b$ is $cos(n_a, n_b)$. Firstly, we calculate the mean inter-class similarity as $S_P$:

$$S_P = \frac{2}{|\mathcal{M}|(|\mathcal{M}|-1)} \sum_{a,b \in \mathcal{M}}^{a \neq b} \frac{1}{|\mathcal{N}_a||\mathcal{N}_b|} \sum_{n_a \in \mathcal{N}_a}^{n_b \in N_b} cos(n_a, n_b). \quad (2)$$

The mean intra-class similarity is formulated by:

$$S_Q = \frac{1}{|\mathcal{M}|} \sum_{c \in \mathcal{M}} \frac{1}{|\mathcal{N}_c|(|\mathcal{N}_c|-1)} \sum_{n_w, n_z \in \mathcal{N}_c}^{w \neq z} cos(n_w, n_z). \quad (3)$$

The formulation of RICD ($D_C$) is as follows:

$$D_C = S_Q - S_P. \quad (4)$$

The high value of $D_C$ means differentiated nodes and a low degree of oversmoothing. The proposed metric takes class differences into consideration, focusing on the relative discrepancy of inter-class nodes compared to intra-class ones, accurately measuring the degree of oversmoothing.

## 3.3 Dimension-Level Decoupling

In the following two sections, we elaborate on the proposed framework: Dual-Dimensional Class-Difference Decoupling (DDCD). It is divided into two components: (1) Dimension-Level Decoupling (DLD in Section 3.3) minimizes covariances between dimensions, reducing the dimensional information redundancy to enhance node differentiation and alleviate oversmoothing. (2) Instance-Level Class-Difference Decoupling (ICDD in Section 3.4), which adaptively attracts or repels intra/inter-class nodes, alleviating oversmoothing while establishing clear class boundaries.

**Motivation.** Existing methods primarily concentrate on instance-level cues related to node relations for mitigating oversmoothing. In the context of self-supervised learning, the dimensional relations have been proven significant in enhancing the representation capacity of the embedding space in prior works [25, 28, 75]. Likewise, we posit that dimensional relations can also enhance the node representation in GNNs to alleviate oversmoothing. Furthermore, in deep GNNs, we observe that the multiple GNN layers lead to a substantially increased information redundancy at the dimensional

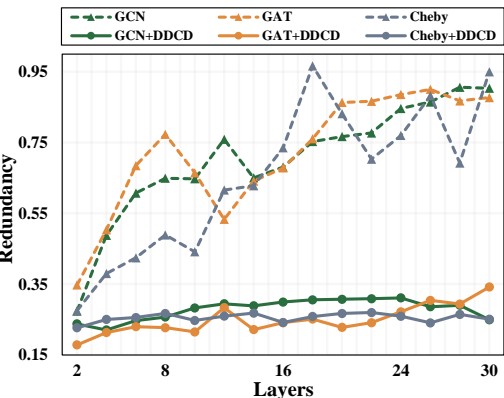

**Figure 3: The Dimensional Information Redundancy on Cora dataset. The results show the escalated dimensional information redundancy with multiple GNN layers, which hinders node differentiation and leads to oversmoothing.**

level, resulting in limited node differentiation and aggravated oversmoothing. Specifically, denote $\mathbf{C}$ as the covariance matrix, which indicates the dimension redundancy of the embedding $\mathbf{z}$:

$$\mathbf{C}_{ij} = \frac{(\mathbf{z}_i - \overline{\mathbf{z}}_i)(\mathbf{z}_j - \overline{\mathbf{z}}_j)}{\sqrt{(\mathbf{z}_i - \overline{\mathbf{z}}_i)^2}\sqrt{(\mathbf{z}_j - \overline{\mathbf{z}}_j)^2}}, \quad (5)$$

where $i$ and $j$ index the dimensions of $\mathbf{z}$, and $\overline{\mathbf{z}}_i$ represents the mean of $\mathbf{z}_i$. As depicted in Figure 3, we compute the mean redundancy across all dimensions of final embeddings. It tends to increase with deeper GNN layers, suggesting a high degree of dimension redundancy. It implies that less information is encoded within the learned dimensions, consequently limiting the representation capacity of node embeddings and the differentiating ability of nodes.

The aforementioned insights motivate us to alleviate oversmoothing in deep GNNs by reducing the dimensional information redundancy, thereby boosting the diversity dimension information and enhancing the node differentiation ability. To extenuate dimensional information redundancy for well-differentiated nodes, we introduce a layer-wise decoupling module, DLD (Dimension-Level Decoupling). Concretely, in a $K$-layer GNN, assume $\mathbf{H}^{(l)}, l \in K$ is the $l$-th layer output, $\mathbf{H}_i^{(l)}$ represents the i-th row of $\mathbf{H}^{(l)}$. We standardize the covariance matrix of the node representation matrix $\mathbf{H}^{(l)}$ and the objective of DLD is defined as follows:

$$\ell_{Dim}(\mathbf{H}^{(l)}) = \sum_i \sum_{j \neq i} \mathbf{C}_{ij}^{(l)2}, \quad (6)$$

where $i, j$ indexes dimensions of node embedding. $\mathbf{C}^{(l)}$ is the auto-covariance matrix of $\mathbf{H}^{(l)}$, which is formulated as:

$$\mathbf{C}_{ij}^{(l)} = \frac{(\mathbf{H}_{:,i}^{(l)} - \overline{\mathbf{H}}_{:,i}^{(l)})(\mathbf{H}_{:,j}^{(l)} - \overline{\mathbf{H}}_{:,j}^{(l)})^T}{\sqrt{(\mathbf{H}_{:,i}^{(l)} - \overline{\mathbf{H}}_{:,i}^{(l)})^2}\sqrt{(\mathbf{H}_{:,j}^{(l)} - \overline{\mathbf{H}}_{:,j}^{(l)})^2}}, \quad (7)$$

$\overline{\mathbf{H}}_{:,i}^{(l)}$ represents the mean value of $\mathbf{H}_{:,i}^{(l)}$. To obtain well-differentiated final embeddings, we apply $\ell_{Dim}$ in each layer, and the loss function $\mathcal{L}_{Dim}$ can be defined as follows:

$$\mathcal{L}_{Dim} = \sum_{l=1}^{K} \ell_{Dim}(\mathbf{H}^{(l)}). \quad (8)$$

Minimizing $\mathcal{L}_{Dim}$ encourages the dimensions independent in each layer, diversifying the dimension information, and ultimately improving the node differentiation for alleviating oversmoothing.

## 3.4 Instance-Level Class-Difference Decoupling

**Motivation.** At the instance level, existing methods alleviate over-smoothing without taking class differences into account, which results in vague classification boundaries. For example, Zhou *et al.* [79] introduced Differentiable Group Normalization (DGN), a normalization method that assigns group probabilities using a learnable linear layer and normalizes node representations based on group relations. However, nodes from different classes can be grouped together, leading to fuzzy class boundaries and degraded performance. To effectively leverage class differences and establish distinct classification boundaries for alleviating oversmoothing, an ideal approach should encourage inter-class nodes to diverge while preserving the similarity of intra-class nodes. Through instance-level decoupling that incorporates class differences, it significantly alleviates the oversmoothing with explicit classification boundaries, enhancing robust node representations.

Inspired by the above discussions, we propose a supervised contrastive learning method Instance-Level Class-Difference Decoupling (ICDD). It leverages class differences to enhance the discriminative capability of nodes at the instance level. The fundamental concept is to minimize the similarity between node representations from different classes while simultaneously preserving the similarity between node representations from the same class. Consequently, the output representations within each layer can be distinctly separated by class, effectively mitigating oversmoothing at the instance level. Denote $\mathcal{N}$ as all node row indexes of the training set, and $\mathcal{S}_i \subseteq \mathcal{N}, i \in \mathcal{N}$ represents node indexes with the same class of i. Denote $\mathcal{D}_i = \mathcal{N} - \{i\}$, for loss function $\ell_{Ins,i}$ of a single node $i$ in $\mathbf{H}^{(l)}$, we have:

$$\ell_{Ins,i} = \frac{-1}{|\mathcal{S}_i|} \sum_{p \in \mathcal{S}_i}^{p \neq i} \log \frac{\exp(\mathbf{H}_i^{(l)} \cdot \mathbf{H}_p^{(l)}/\tau)}{\sum_{q \in \mathcal{D}_i} \exp(\mathbf{H}_i^{(l)} \cdot \mathbf{H}_q^{(l)}/\tau)}. \quad (9)$$

For nodes in the training set, the formulation is as follows:

$$\ell_{Ins}(\mathbf{H}^{(l)}) = \frac{1}{|\mathcal{N}|} \sum_{i \in \mathcal{N}} \ell_{Ins,i}. \quad (10)$$

Applying $\ell_{Ins}$ in each GNN layer, $\mathcal{L}_{Ins}$ is as follows:

$$\mathcal{L}_{Ins} = \sum_{l=1}^{K} \ell_{Ins}(\mathbf{H}^{(l)}). \quad (11)$$

Minimizing $\mathcal{L}_{Ins}$ prompts the representations of each layer to be discriminative while maintaining the similarity of nodes from the same class, thereby mitigating the oversmoothing issue with explicit classification boundaries.

## 3.5 Overall Objective Function

The dimensional decoupling loss in Equation (8) diversifies dimension information and enhances the dimension-level node differentiation, while the instance-level decoupling loss in Equation (11) promotes well-differentiated nodes with explicit class boundaries. Besides, the typical classification loss is utilized to provide supervision for the classification task. Denote $\mathcal{L}_0$ as the classification

| Dataset | Nodes | Edges | Features | Classes |
|---------|-------|-------|----------|---------|
| Cora | 2708 | 5429 | 1433 | 7 |
| Citeseer | 3327 | 4732 | 3703 | 6 |
| Pubmed | 19717 | 44338 | 500 | 3 |
| CoauthorCS | 18333 | 81894 | 6805 | 15 |
| ogbn-arxiv | 169343 | 1166243 | 128 | 40 |

**Table 1: Dataset Statistics.**

loss, and $\ell_{ce}(v_i, y_i)$ as calculating the cross-entropy loss with the representation of node $v_i$ and its class labels $y_i$. The classification loss can be formulated as follows:

$$\mathcal{L}_0 = \frac{1}{|\mathcal{V}_s|} \sum_{v_i \in \mathcal{V}_s} \ell_{ce}(v_i, y_i). \quad (12)$$

Combine with Equation (8), Equation (11), Equation (12), the overall objective function can be stated as (where $\alpha > 0$ is a coefficient that balances the trade-off of dimension decoupling):

$$\begin{aligned} \mathcal{L} &= \alpha \mathcal{L}_{Dim} + \mathcal{L}_{Ins} + \mathcal{L}_0 \\ &= \alpha \sum_{l=1}^{K} \ell_{Dim}(\mathbf{H}^{(l)}) + \sum_{l=1}^{K} \ell_{Ins}(\mathbf{H}^{(l)}) + \frac{1}{|\mathcal{V}_s|} \sum_{v_i \in \mathcal{V}_s} \ell_{ce}(v_i, y_i). \end{aligned} \quad (13)$$

Compared to existing methods that primarily focus on instance-level relations, we reveal the intensified dimension redundancy in deep GNNs and introduce a novel approach to eliminate the redundancy for enhancing node differentiation. Additionally, we propose utilizing class differences in the instance-level decoupling, alleviating oversmoothing with explicit classification boundaries. The dual-dimensional decoupling strategies complement each other and further enhance the overall performance.

## 4 Experiment

**Datasets.** We conducted experiments on five well-known benchmarks: Cora, Citeseer, Pubmed [69], CoauthorCS [53] and ogbn-arxiv [24]. For the experiments on Cora/Citeseer/Pubmed, we followed the semi-supervised setting proposed by Kipf *et al.* [32], with 20 nodes per class for training. For CoauthorCS, we adopted the same setting in [79], using 40 nodes per class for training, 150 nodes per class for validation, and the remaining for testing. Besides, we also conduct experiments on large-scale datasets such as ogbn-arxiv [24], to demonstrate the effectiveness of DDCD to data scalability. The detailed statistics of datasets can be found in Table 1.

**Baselines.** We conducted experiments on three GNN backbones: GCN [32], GAT [57], and ChebyNet [12]. We compared three regularization baseline methods: BatchNorm (BN) [27], PairNorm (PN) [78], DGN [79], as well as three dropout methods: DropEdge (DE) [50], SkipNode (SN) [42], DropMessage (DM) [15]. To prove the effectiveness of DDCD in boosting deep GNNs, we experimented on two deep GNNs: GCNII [44] and EGNN [80].

**Hyperparameters.** We adopted the same experimental setting as DGN [79]. For Cora, Citeseer, and CoauthorCS, we used a dropout rate of 0.6, L2 regularization of $5 \cdot 10^{-4}$, and a learning rate of $5 \cdot 10^{-3}$. For Pubmed, we used a dropout rate of 0.6, L2 regularization of $1 \cdot 10^{-3}$, and a learning rate of $1 \cdot 10^{-2}$. We conducted each experiment five times and reported the average results. For BatchNorm, PairNorm, and DGN, we followed the results reported by Zhou *et*

| Dataset | Layer | GCN | | | | | | | | GAT | | | | | | | | Cheby | | | | | | | |
|---|---|---|---|---|---|---|---|---|---|---|---|---|---|---|---|---|---|---|---|---|---|---|---|---|---|
| | | NN | DE | SN | DM | BN | PN | DGN | DDCD | NN | DE | SN | DM | BN | PN | DGN | DDCD | NN | DE | SN | DM | BN | PN | DGN | DDCD |
| Cora | 2L | **82.20** | 79.94 | 79.41 | 81.15 | 73.90 | 71.00 | 82.00 | 80.36 | 80.90 | **81.44** | 78.25 | 79.93 | 77.80 | 71.00 | 81.10 | 79.72 | **81.40** | 79.56 | 78.56 | 79.63 | 67.92 | 65.52 | 79.98 | 79.70 |
| | 15L | 18.10 | 45.02 | 63.12 | 63.89 | 70.30 | 67.20 | 75.20 | **77.22**↑2.02 | 16.80 | 44.70 | 52.37 | 53.46 | 33.10 | 49.60 | 71.80 | **73.70**↑1.90 | 30.62 | 49.50 | 62.24 | 63.41 | 68.30 | 60.28 | 76.80 | **77.20**↑0.40 |
| | 30L | 13.10 | 27.60 | 57.92 | 59.43 | 67.20 | 64.30 | 73.20 | **73.92**↑0.43 | 13.00 | 24.42 | 38.59 | 40.57 | 25.00 | 30.20 | 51.30 | **56.33**↑5.03 | 30.40 | 31.38 | 49.83 | 50.29 | 60.30 | 52.54 | 66.20 | **69.00**↑2.80 |
| Citeseer | 2L | **70.60** | 68.45 | 67.83 | 68.96 | 51.30 | 60.50 | 69.50 | 69.91 | 70.20 | 67.30 | 67.39 | 68.25 | 61.50 | 62.00 | 69.30 | **71.20**↑1.90 | 67.78 | 67.12 | 65.23 | 64.46 | 47.50 | 52.40 | 65.66 | 67.20 |
| | 15L | 15.20 | 25.80 | 39.42 | 42.37 | 46.90 | 46.70 | 53.10 | **59.90**↑6.80 | 22.60 | 21.66 | 30.67 | 38.53 | 28.00 | 41.40 | 52.60 | **53.10**↑0.50 | 44.78 | 42.28 | 40.62 | 41.79 | 40.60 | 40.88 | 48.50 | **56.60**↑8.10 |
| | 30L | 9.40 | 12.62 | 32.69 | 38.54 | 47.90 | 47.10 | 52.60 | **56.12**↑3.52 | 7.70 | 7.74 | 12.73 | 28.66 | 21.40 | 33.30 | 45.60 | **46.20**↑0.60 | 35.90 | 28.46 | 28.33 | 30.67 | 36.88 | 38.12 | 45.68 | **50.00**↑4.32 |
| Pubmed | 2L | **79.30** | 77.56 | 74.32 | 78.47 | 74.90 | 71.10 | 78.23 | 78.50 | **77.80** | 77.28 | 77.16 | 77.45 | 76.20 | 72.40 | 77.50 | 76.60 | 78.48 | 77.86 | 75.43 | 78.15 | 73.69 | 72.95 | 78.42 | 78.30 |
| | 15L | 22.50 | 34.24 | 57.63 | 62.57 | 73.70 | 70.60 | 76.10 | **77.70**↑1.60 | 37.50 | 36.88 | 59.33 | 61.64 | 56.20 | 68.80 | 75.90 | **78.40**↑2.50 | 46.36 | 44.66 | 59.62 | 61.03 | 70.22 | 69.16 | 76.08 | **77.50**↑1.42 |
| | 30L | 18.00 | 22.56 | 21.98 | 37.43 | 70.40 | 76.90 | 76.90 | **77.10**↑0.20 | 18.00 | 18.31 | 23.55 | 38.93 | 46.60 | 58.20 | 73.30 | **74.60**↑1.30 | 39.58 | 40.52 | 57.23 | 59.06 | 71.14 | 57.74 | 70.24 | **74.20**↑3.06 |
| Coauthor. | 2L | 92.30 | 91.93 | 91.86 | 92.34 | 86.00 | 77.80 | 92.30 | **92.70**↑0.36 | 91.50 | 90.63 | 90.42 | 91.67 | 89.40 | 85.90 | 91.80 | **91.92**↑0.12 | **93.01** | 93.10 | 92.79 | 93.24 | 81.95 | 72.95 | 92.29 | 92.95 |
| | 15L | 72.20 | 37.61 | 45.63 | 52.13 | 78.50 | 69.50 | 83.70 | **85.34**↑1.64 | 6.00 | 12.68 | 13.75 | 13.96 | 77.70 | 53.10 | 84.50 | **86.44**↑1.94 | 70.04 | 35.02 | 48.77 | 53.83 | 76.81 | 57.74 | 83.36 | **89.03**↑5.67 |
| | 30L | 3.30 | 26.53 | 34.32 | 40.71 | 84.30 | 64.50 | 84.40 | **84.48**↑0.08 | 3.30 | 10.69 | 12.33 | 12.79 | 16.70 | 48.10 | 75.50 | **76.30**↑0.80 | 35.18 | 20.33 | 31.44 | 34.73 | 81.77 | 65.80 | 79.04 | **82.58**↑0.81 |

**Table 2: The Node Classification Accuracy with 2, 15, and 30-layer GNNs. NN: vanilla GNN, DE: DropEdge [50], SN: SkipNode [42], DM: DropMessage [15], BN: BatchNorm [27], PN: PairNorm [78], DGN: Differentiable Group Normalization [79]. DDCD: Our method. Bold represents the highest accuracy. Underline represents the second-highest accuracy. See details in Section 4.1.**

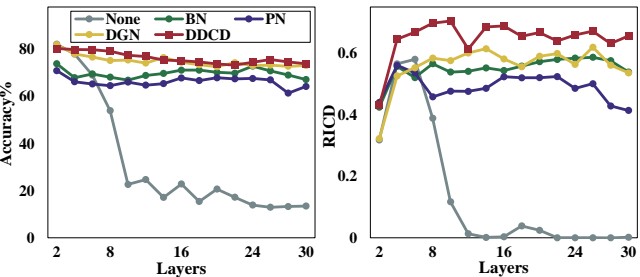

**Figure 4: The Node Classification Accuracy and Relative Inter-Class Distance Comparison equipped with different regularization methods. Our method outperforms the baselines in different layer cases. Refer to Section 4.1 for more details.**

*al.* [79] on GCN and GAT. To investigate the effects of hyperparameters, we adjusted the hyperparameters $\alpha$ in {0.002, 0.004, 0.006, 0.008, 0.01} and $\tau$ in {0.01, 0.03, 0.05, 0.07, 0.09} on the Cora dataset, reported the best performance with $\alpha = 0.006$ and $\tau = 0.05$. Then we fixed $\tau$, adjusted $\alpha$ in {$6 \cdot 10^{-6}$, $6 \cdot 10^{-5}$, $6 \cdot 10^{-4}$, $6 \cdot 10^{-3}$} on Citeseer, Pubmed, and CoauthorCS, reported the best performance and recorded the values of $\alpha$ respectively: $6 \cdot 10^{-5}$ on Citeseer, $6 \cdot 10^{-4}$ on Pubmed and CoauthorCS.

## 4.1 Comparison with State-of-the-Art Methods

We provide comparison results on both regularization and dropout methods with different backbones and layer numbers. Firstly, we compare the baseline methods with 2, 15, and 30 layers on different GNNs. As shown in Table 2, compared with vanilla GNNs, DDCD significantly alleviates performance degradation in deep GNNs; Compared with other baseline methods, DDCD can achieve the highest classification performance in the same number of layers, which means better discrimination of node features, i.e., mitigating oversmoothing better. Subsequently, we further provided comparison results with different regularization methods in different layer cases, reporting the node classification accuracy of GCN on the Cora dataset, for a range of layer numbers in {2, 4, ..., 30}. As shown in Figure 4, our proposed method achieved the best classification accuracy across various layer numbers, indicating that it performs better at mitigating oversmoothing in most cases. Additionally, to

| Component | | Cora | | | | | | |
|---|---|---|---|---|---|---|---|---|
| DLD | ICDD | 2L | 10L | 15L | 20L | 30L | AVG | △ |
| | | **82.20** | 22.68 | 18.10 | 20.68 | 13.10 | 31.32 | - |
| ✓ | | 81.76 | 62.48 | 48.36 | 27.78 | 27.52 | 49.58 | ↑18.26 |
| | ✓ | 81.76 | 67.24 | 68.84 | 65.08 | 40.50 | 64.68 | ↑33.36 |
| ✓ | ✓ | 80.36 | **77.26** | **77.22** | **74.32** | **73.92** | **76.62** | ↑45.30 |

**Table 3: Ablation Study with DLD and ICDD. The precise accuracy values are reported. See Section 4.2 for details.**

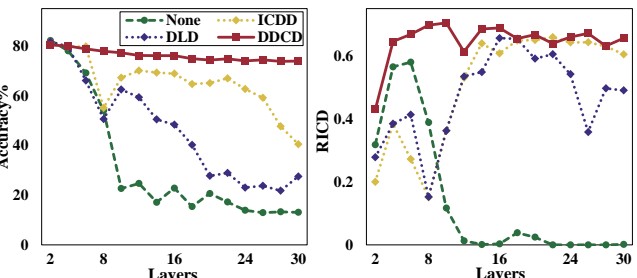

**Figure 5: Ablation Study with DLD and ICDD. The node classification accuracy and Relative Inter-Class Distance are reported. Please refer to Section 4.2 for more details.**

quantitatively assess the performance of alleviating oversmoothing, we reported the Relative Inter-Class Distance (RICD) on the Cora dataset with various layers on the GCN model. Our method consistently exhibited the largest RICD at all layer cases, indicating the lowest degree of oversmoothing. All results confirm that our method achieves superior mitigation compared to baseline methods on different GNN models and various network layers.

## 4.2 Ablation Study

Firstly, we investigated the influence of two components, Dimension-Level Decoupling (in Section 3.3) and Instance-Level Class-Difference Decoupling (in Section 3.4), on the performance of the proposed method. We empirically demonstrate that both components have a positive impact on alleviating oversmoothing and can complementarily boost performance. Next, we investigate the effects of hyperparameters, demonstrating the stability with different parameters. Besides, we provide comparisons between Relative Inter-Class

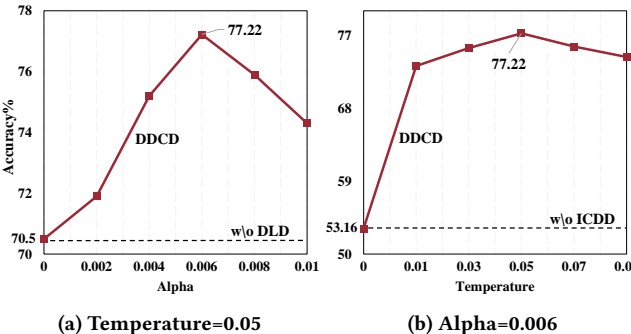

**(a) Temperature=0.05**    **(b) Alpha=0.006**

Figure 6: Ablation Study on Hyperparameters. (a) reports the ablation with different alpha $\alpha$, and (b) reports the ablation with different temperatures $\tau$. The results show the stability of our method to hyperparameters. See details in Section 4.2.

Distance (RICD) and existing metrics for oversmoothing, demonstrating the superiority of the proposed measurement.

**Effect of Each Component.** To thoroughly assess the effectiveness of DDCD, we conducted extensive ablation experiments on the Cora dataset with varying numbers of GCN layers. Based on the results presented in Table 3 and Figure 5, two main conclusions can be drawn. **Firstly,** both components have a positive impact on mitigating oversmoothing and enhancing performance. As depicted in Figure 5, both DLD and ICDD result in a larger RICD compared to the baseline, signifying the effective mitigation of oversmoothing; Additionally, as shown in Table 3, the utilization of DLD and ICDD results in an average performance improvement of 18.26% and 33.36% respectively, further indicating the successful alleviation of oversmoothing, subsequently enhancing classification performance across various layer scenarios. **Secondly,** the combination of these two components can complement each other and further enhance the performance. As illustrated in Table 3, the combination of DLD and ICDD outperforms the baseline by an average of 45.30%, underscoring the collaborative effect of both DLD and ICDD.

**Effect of Hyperparameters.** To investigate the impact of hyperparameters $\tau$ and $\alpha$ for the proposed method, we conducted two ablation experiments on the Cora dataset with 15-layer GCN. Specifically, we performed the following experiments with DDCD and reported the node classification accuracy: (1) we changed the value of $\alpha$ while keeping $\tau$ fixed at 0.05, and (2) changed the value of temperature $\tau$ while keeping the value of $\alpha$ fixed at 0.006.

In Experiment (1), as illustrated in Figure 6 (b), the performance of DDCD remains stable with changes in $\alpha$, and is further boosted with Dimension-Level Decoupling equiped; In Experiment (2), as shown in Figure 6 (a), we observed that the performance of DDCD remains stable with different values of $\tau$, and is further improved with ICDD equiped. Specifically, on the Cora dataset, our proposed method achieves the best performance with $\tau = 0.05$. We then apply the same value of $\tau$ on the Citeseer, Pubmed, and CoauthorCS datasets, and obtain similarly strong results. Both Experiment (1) and (2) show the stability of our method to hyperparameters, and the complementary of two modules.

**Pair-wise Distance v.s. Relative Inter-Class Distance.** The pair-wise distance measures the degree of oversmoothing based on the distance between node pairs [6, 23, 40]. However, the pair-wise

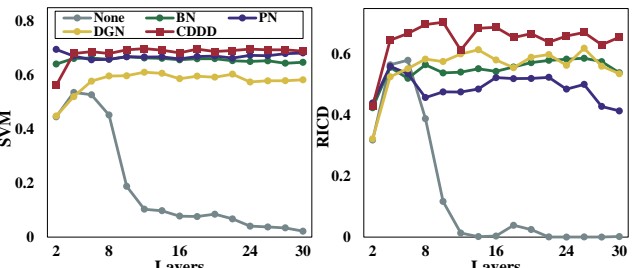

Figure 7: Metric Comparisons. The SMV [40] and Relative Inter-Class Distance (RICD) on the Cora dataset are reported. RICD can accurately measure oversmoothing compared to existing metrics. Please refer to Section 4.2.

distances are closely tied to graph information, such as class labels. In the context of the node classification task, nodes in distinct classes should display discernible pair-wise distances, while nodes within the same class should exhibit closer pair-wise distances. Failure to account for class information in the computation of pair-wise distances results in an impact from class labels, leading to an inaccurate assessment of oversmoothing. Specifically, we provide the comparisons of Relative Inter-Class Distance and the existing metric SMV from [40] on the GCN model. As depicted in Figure 7, SMV fails to accurately reflect the superiority of different methods. Notably, PairNorm and BatchNorm cannot be distinguished in SMV, and DGN demonstrates better oversmoothing alleviation but lower SMV than PN and BN. In contrast, the proposed RICD accurately reflects the superiority of different methods and the degree of oversmoothing alleviation.

## 4.3 Generalization Analysis

We conducted comprehensive experiments to prove the generalization of our method in various scenarios. **First,** we conducted experiments with SOTA deep GNN models, to emphasize the effectiveness of our approach in enhancing deep GNNs as a regularization technique. **Additionally,** we conducted experiments to demonstrate the robustness of our method in different settings and layer numbers. **Moreover,** we visualized the learned node representations, which depicts that DDCD is feasible to learn discriminative features for different classes.

**Boosting Deep GNN Models.** Our proposed DDCD can be treated as a regularization for addressing oversmoothing, which can be seamlessly incorporated into different deep GNN models. To demonstrate the effectiveness of our method in enhancing deep GNNs, we apply DDCD to two deep models [44, 80] for graph data, with various network layers on the Cora dataset. As demonstrated in Table 4, our method significantly enhances the performance of deep GNNs by mitigating the oversmoothing issue.

**Generalization to Large-Scale Graph.** To demonstrate the generalization of the proposed DDCD to the graph scalability, we conduct experiments on the large-scale graph: ogbn-arxiv. To reduce the computational complexity, we randomly sample $\sqrt{N}$ nodes to apply DDCD in each GNN layer. As shown in Table 5, we perform experiments on GCN and GAT with different layers, and the performance is consistently improved across all cases, demonstrating the robustness of DDCD to dataset scalability.

| Method | GCNII [44] | | | EGNN [80] | | |
|---|---|---|---|---|---|---|
| | 2L | 32L | 64L | 2L | 32L | 64L |
| Base | 82.40 | 84.80 | 85.40 | 83.20 | 85.50 | 85.70 |
| + DDCD | **82.60** | **85.60** | **86.30** | **83.80** | **85.67** | **86.10** |

**Table 4: Experiments with Deep GNNs. Classification accuracy on the Cora Dataset is reported, indicating the effectiveness of DDCD for alleviating oversmoothing in deep GNNs.**

| Method | GCN | | | GAT | | |
|---|---|---|---|---|---|---|
| | 2L | 15L | 30L | 2L | 15L | 30L |
| Base | 70.40 | 70.60 | 68.50 | 68.40 | 72.70 | 72.70 |
| + DDCD | **70.60** | **70.84** | **68.92** | **68.80** | **72.95** | **72.95** |

**Table 5: Experiments on Large-Scale Graph. Classification accuracy on the ogbn-arxiv dataset is reported. The results show DDCD can generalize to large-scale graphs.**

| Method | GCN | | | | | | |
|---|---|---|---|---|---|---|---|
| | 2L | 8L | 10L | 15L | 20L | 25L | 30L |
| Base | 71.79 | 35.78 | 12.44 | 8.49 | 8.35 | 8.36 | 8.35 |
| + DDCD | **71.81** | **74.88** | **77.15** | **77.19** | **76.96** | **74.47** | **72.07** |

**Table 6: Experiments on Imbalanced Classes. Classification accuracy on the Cora dataset is reported. The results prove the generalization of DDCD with imbalanced classes.**

**Generalization to Imbalanced Classes.** We leveraged class differences to address oversmoothing, and the performance of our method may potentially be influenced by the imbalanced numbers of nodes among different classes. To investigate the generalization of DDCD to the imbalanced classes, we re-sampled the Cora dataset, where one of the classes has three times as many nodes as the other classes. As depicted in Table 6, our method consistently and effectively mitigates oversmoothing in all layer cases, indicating the robustness of DDCD to imbalanced classes.

**Different Numbers of Layers with GAT/Cheby.** To demonstrate the generalization of DDCD with various layer numbers, we provided experimental results for the GAT and Cheby models with different numbers of layers. As depicted in Figure 8, we experimented from 2 to 30 layers, and the incorporation of DDCD leads to a larger RICD and consistently improves classification accuracy in all layer scenarios, signifying the effective alleviation of oversmoothing and consequently boosting performance.

**Node Representation Visualization.** We visualized the node representations of vanilla GCN, DGN [79], and DDCD with a 30-layer model on the Cora dataset. As depicted in Figure 9 (a), when with deep network architecture (30 layers), the node representations become mixed and cannot be clearly distinguished, indicating severe oversmoothing and leading to a decline in classification performance. Upon the integration of DGN [79], as illustrated in Figure 9 (b), nodes can be roughly categorized by class, but the inappropriate grouping strategy blurs the nodes at the classification

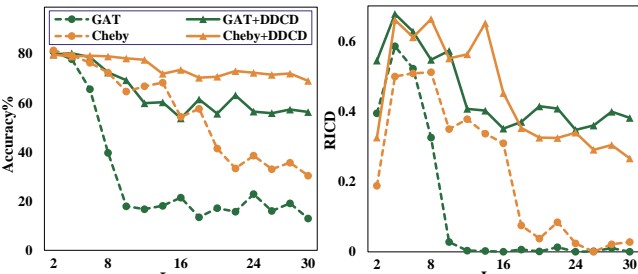

**Figure 8: More Layer Cases with GAT/Cheby. The classification accuracy and Relative Inter-Class Distance on the Cora dataset are reported. The results demonstrate the generalization of DDCD with different layers.**

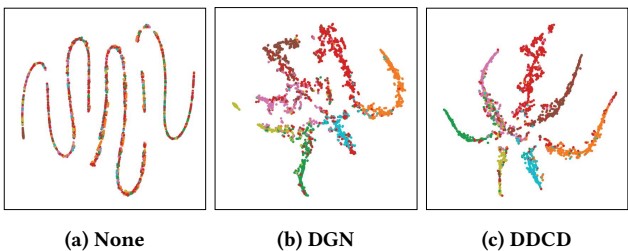

(a) None                (b) DGN                (c) DDCD

**Figure 9: Visualization of Node Representations: (a), (b), and (c) visualized node embeddings with 30-layer vanilla GCN and the one equipped with DGN [79]/DDCD. The result demonstrates DDCD promotes discriminative features effectively. Colors indicate nodes from different classes.**

boundaries. However, as shown in Figure 9 (c), the node representations exhibit well discrimination and explicit classification boundaries equipped with DDCD.

## 5 Conclusion

In deep GNNs, node representations become highly indistinguishable due to repeated aggregations, i.e., the oversmoothing issue. In this paper, we construct a simple and effective dual-dimensional regularization method to address the oversmoothing issue. Specifically, we consider the oversmoothing issue from two aspects of the node embedding space: dimension and instance. At the dimension level, we first reveal the dimensional causes for oversmoothing and propose to alleviate oversmoothing by minimizing covariances between dimensions to enhance node differentiation. Besides, at the instance level, we propose to utilize class differences, alleviating oversmoothing with explicit class boundaries. This research provides valuable insights for tacking oversmoothing in deep GNNs and will facilitate potential real-world applications. In future work, it is significant to generalize our instance-level method for tasks lacking node labels, e.g., link prediction tasks, where linked/unlinked nodes can be regarded as intra/inter-class samples and instance-level decoupling helps obtain discriminative node representations.

# Acknowledgments

This work is supported by National Natural Science Foundation of China under Grant (62176188, 62225113, 623B2080). The numerical calculations in this paper have been supported by the supercomputing system in the Supercomputing Center of Wuhan University.

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
