# OpenReview forum: "Resisting Over-Smoothing in Graph Neural Networks via Dual-Dimensional Decoupling"
_acmmm.org/ACMMM/2024/Conference — MM2024 Poster_

### Official Review · Reviewer_jMg7 · 2024-05-20

**Rating:** 5
**Confidence:** 4

**Summary:**

This paper proposed Dual-Dimensional Class-Difference Decoupling (DDCD), which addresses the oversmoothing issue in Graph Neural Networks (GNNs) by reducing dimension redundancy with Dimension-Level Decoupling (DLD) and enhancing node differentiation by considering class differences with Instance-Level Class-Difference Decoupling (ICDD), supported by a novel evaluation metric.

**Strengths:**

1. It's interesting and holds novelty in tracking over-smoothing problems from the node feature dimension.
2. The motivation for using "Dimenstion-level Decoupling" and "instance-level class-difference Decoupling" is clear and convincing.
3. A new metric for similarity between nodes is proposed.
4. The organization and writing are good and easy to follow, and every figure supports the claim of the authors.
5. The detailed and comprehensive experiments show the effectiveness of the proposed method.

**Limitations:**

1. It's better to compare with some SOTA GCN methods, especially after 2023, to show the effectiveness of the proposed method.
2. For the proposed metric, Relative Inter-Class Distance (RICD), if one sets the group number the same as the class number for DGN, what is the difference between it and RICD?
3. Some recent research about over-smoothing problems is missing, please add some discussion:
   - Duan, W., J. Xuan, M. Qiao, and J. Lu. “Learning from the Dark: Boosting Graph Convolutional Neural Networks With Diverse Negative Samples”. Proceedings of the AAAI Conference on Artificial Intelligence, vol. 36, no. 6, June 2022, pp. 6550-8.
   - Wei Duan, Jie Lu, Yu Guang Wang, & Junyu Xuan (2024). Layer-diverse Negative Sampling for Graph Neural Networks. Transactions on Machine Learning Research.

**Suitability:**

2

---

### Official Review · Reviewer_scqz · 2024-05-20

**Rating:** 3
**Confidence:** 4

**Summary:**

This study addresses the oversmoothing issue in GNNs by proposing a approach with two modules. Dimension-Level Decoupling (DLD) targets redundancy in node dimensions, improving differentiation, while Instance-Level Class-Difference Decoupling (ICDD) focuses on enhancing classification clarity by managing inter-class and intra-class node distinctions. Extensive experiments validate the effectiveness of the proposed Dual-Dimensional Class-Difference Decoupling (DDCD) across various scenarios, demonstrating its potential to significantly enhance GNN performance.

**Strengths:**

- This paper is well-organized, narrating from the background and existing problems to their solutions.
- The paper features excellent images and well-formatted tables, experiments are comprehensive.
- I appreciate the authors for providing code, which is easy to read.

**Limitations:**

The main concerns revolve around the novelty and rationality of the proposed solutions:
- The proposed method serves as a normalization approach to tackle the over-smoothing problem. However, simpler methods such as initial residual [1, 2], combining hidden representations from multiple hops/layers [3, 4], and decoupling aggregation and transformation [5] have already addressed this issue using straightforward strategies, enabling GNNs to stack hundreds of layers without performance degradation. Notably, the experiments incorporate several backbones but yield minimal gains; a 2-layer GCN often achieves the best performance in most situations. Therefore, the necessity of normalization tricks is questionable.
- DLD aims to minimize the non-diagonal elements of the covariance matrix in each layer. However, this could be achieved with traditional methods like PCA. Using it as a loss function may be ok but somewhat weird, at least a further disscusion is needed.
- The use of contrastive loss to over-smoothing has been previously explored [6]. The authors fail to discuss how their approach differs from this. Additionally, there is a significant discrepancy between the results reported in their paper and those in ContraNorm [6]; could the authors clarify this?
In summary, while the paper exhibits excellent research skills, the problems it addresses and the solutions it proposes lack significant impact.

[1] Predict then propagate: Graph neural networks meet personalized pagerank

[2] Simple and deep graph convolutional networks

[3] Adagcn: Adaboosting graph convolutional networks into deep models

[4] Representation learning on graphs with jumping knowledge networks

[5] Dissecting the diffusion process in linear graph convolutional networks

[6] Contranorm: A contrastive learning perspective on oversmoothing and beyond

**Suitability:**

3

---

### Official Review · Reviewer_8kPW · 2024-05-23

**Rating:** 4
**Confidence:** 4

**Summary:**

The paper introduces a novel dual-dimensional decoupling mechanism, which focuses on the over-smoothing problem in graph neural networks. It tackles the main challenge: most existing methods alleviate the over-smoothing problem at the instance level while ignoring the dimension-level node differentiation. The authors adopt a dual-dimensional decoupling mechanism to reduce the redundant information within node embedding dimensions. Besides, a novel metric to measure over-smoothing is designed. The experimental results on several real-world datasets demonstrate the effectiveness of the proposed model.

**Strengths:**

1. The paper is well-motivated since over-smoothing is a significant problem in graph representation learning. Besides, dimension-oriented mechanism to reduce the redundant information is interesting and worth exploring.

2. The paper provides a well-structured overview, making it easy to follow for readers.

3. It encompasses all the essential components required in tackling the over-smoothing problem, offering a highly comprehensive coverage.

4. The runnable code is provided, making readers easy for reproduction of experimental results.

5. A vast set of experiments were conducted in several benchmark datasets, showing the superiority of the proposed method.

**Limitations:**

1. There seem to lack analysis about how previous researches addressed the over-smoothing problem with dimensional redundant.

2. There seem to lack some SOTA compared methods with decoupling, such as [1].

[1] Zhang, Hongyuan, Yanan Zhu, and Xuelong Li. "Decouple Graph Neural Networks: Train Multiple Simple GNNs Simultaneously Instead of One." IEEE Transactions on Pattern Analysis and Machine Intelligence (2024).

3. There is no definition of decoupling, and it is not clear why the measurement with covariance matrix can be seen as a decoupled way.

4. I am not sure whether this kind of work for single graph is consistent with the multimedia theme of ACM MM 2024.

**Suitability:**

2

---

### Official Review · Reviewer_FFDe · 2024-05-27

**Rating:** 4
**Confidence:** 3

**Summary:**

This work addresses the over-smoothing issue in Graph Neural Networks (GNNs), where deep GNNs suffer from indistinguishable node representations due to repeated aggregations. By approaching the problem from two perspectives: the dimension and the instance level. And introducing Dimension-Level Decoupling (DLD) to reduce dimensional information redundancy, thereby enhancing node differentiation. Additionally, the proposed Instance-Level Class-Difference Decoupling (ICDD) to improve instance-level node discrimination by repelling inter-class nodes and attracting intra-class nodes, resulting in clearer classification boundaries. This work also presents a novel evaluation metric that considers class differences' impact on node distances, facilitating precise over-smoothing measurement. Extensive experiments demonstrate the effectiveness of the proposed Dual-Dimensional Class-Difference Decoupling (DDCD) method across various scenarios.

**Strengths:**

1. This work introduces a novel dual-dimensional decoupling approach to address the over-smoothing problem in deep Graph Neural Networks (GNNs). This method is adaptable to various GNN architectures.
2. The work incorporates class differences at the instance level to alleviate over-smoothing by establishing explicit class boundaries.
3. The work introduces a new metric to measure over-smoothing. This metric quantifies the similarity differences between inter-class and intra-class nodes, taking into account the impact of class differences, thereby providing an accurate measure of over-smoothing extent.
4. The paper is well-organized and clearly written, it’s easy to follow.

**Limitations:**

1. Effectiveness of Deep GNNs: The necessity of deep GNNs is questionable. The results in Table 2 indicate that the performance of DDCD is not as effective as a 2-layer GNN, raising concerns about the benefits of using deeper architectures.
2. Misalignment with Contrastive Learning: Although the RELATED WORK section highlights a focus on contrastive learning on graphs, the paper lacks a detailed analysis of how the proposed method aligns with contrastive learning principles. Additionally, the experimental section does not include baselines from graph contrastive learning, which is a significant omission.
3. Advanced Baselines: The paper compares DDCD with several baselines, but these baselines are not recent or advanced methods. It is crucial to include more current and sophisticated baselines, such as UAG[1], to provide a more comprehensive evaluation of the proposed method's effectiveness.

[1]Kong H, Kim S, Kim H J, et al. Unknown-Aware Graph Regularization for Robust Semi-supervised Learning from Uncurated Data[C]//Proceedings of the AAAI Conference on Artificial Intelligence. 2024, 38(12): 13265-13273.

**Suitability:**

2

---

### Meta-Review · Area_Chair_AR1B · 2024-06-30

**Recommendation:** Accept (Poster)
**Confidence:** 2

**Metareview:**

This paper addresses the over-smoothing issue in Graph Neural Networks. The overall score is positive (5 4 4 3). In the rebuttal phase, reviewer scqz raised the score from 2 to 3 but still has concerns about over-smoothing which has been solved by other works.

To conclude, this paper is a good one even though there exist several limitations.